# Exploring the Digital Divide among the Bhutanese Refugee Community during COVID-19: Engaged Research in Action

**DOI:** 10.3390/ijerph192416854

**Published:** 2022-12-15

**Authors:** Jeffrey H. Cohen, Arati Maleku, Sudarshan Pyakurel, Taku Suzuki, Shambika Raut, Francisco Alejandro Montiel Ishino

**Affiliations:** 1Department of Anthropology, The Ohio State University, Columbus, OH 43210, USA; 2College of Social Work, The Ohio State University, Columbus, OH 43210, USA; 3The Bhutanese Community of Central Ohio, Columbus, OH 43229, USA; 4International Studies, Denison University, Granville, OH 43023, USA; 5National Institute of Environmental Health Sciences, National Institutes of Health, Bethesda, MD 20892, USA

**Keywords:** digital divide, Bhutanese refugees, health inequalities, engaged community research, COVID-19, mixed-methods

## Abstract

The digital divide proved a critical barrier to accessing information and healthcare during the COVID-19 pandemic and negatively impacted the Bhutanese refugee community. Moving beyond a technological model of the digital divide that highlights a lack of access to computers and the internet, we engaged the community to co-produce a dynamic approach that identifies the impact of socio-cultural and socio-environmental factors as well. Our paper reports on our community-academic research partnership and explores how the digital divide exacerbates health disparities in a midwestern Bhutanese refugee community. Combining the efforts of the community, anthropologists and social workers, this paper reports on the health disparities that confront the community as well as interventions designed to mitigate social inequities.

## 1. Introduction

The digital divide, or the gap between those who have affordable access, skills, and support to effectively engage online and those who do not [1] is a critical factor increasing social inequalities around healthcare and access to information on the COVID-19 pandemic [2,3]. For many immigrant and refugee communities, access to information and technology remains a complex challenge that is influenced by social history, prejudice, and assumptions of “natural” limitations on their ability and experiences.

Though higher incomes can facilitate broadband access, the digital divide cannot be bridged through programs that assume income inequality is its sole cause. The digital divide is dynamic and in addition to limits on broadband access and technology [4,5], it is defined by cultural practices, history, environment, and traditions. The digital divide creates social isolation that is intensified by language barriers, lack of training in how to best access and use digital media [6] and a lack of experience and engagement with technologies that often manifests as a generalized mistrust of the internet [7]. These forces are accentuated for minoritized communities as they are further isolated by structural marginalities that perpetuate social, economic, and political disparities [3].

The combination of the many forces influencing access makes it difficult to model solutions and bridge the digital divide. Putting technologies in place that expand broadband access, encouraging income growth and promoting competition among providers can succeed at increasing access; however, these are not magic bullets. Bridging the digital divide demands that we foster culturally grounded solutions that acknowledge the forces, both positive and negative, that impact a population’s use and engagement with technology and increase access to, and utilization of, broadband technologies [8,9]. Solutions that acknowledge the costs and support positive engagement with technology to resolve challenges like the digital divide must include the community. Working with the Bhutanese refugee community we co-founded a solution that recognizes the dynamics of the digital divide and models a sustainable solution that can serve it moving forward. Our paper begins with a brief introduction to the Bhutanese refugee community and critical reflection on our project. This is followed by a discussion of our methodology and highlights the importance of coproduction of shared knowledge and solutions. Next, we share data and capture the dynamic nature of the digital divide for the Bhutanese refugee community in the US. In our conclusions, we argue that bridging the digital divide demands we rethink its meaning and complexity as we partner with communities to define and co-create solutions.

### 1.1. The Bhutanese Refugee Community

The Bhutanese refugee community in the US is unique. Hailing from South Asia, Bhutanese refugees are part of a large resettlement program coordinated by the UNHCR [10]. Over 86,000 Bhutanese refugees have relocated to the US since 2006 with over 30,000 settled in the Midwest, the setting of our study [11].

Approximately one-third of the Bhutanese population lives in poverty, more than double the national poverty average. While South Asians are often celebrated for their educational success, only about 9% of US Bhutanese have completed a bachelor’s degree by 25 years of age. Additionally, the Bhutanese are at risk for high blood pressure, obesity and cardio-metabolic diseases [12]. Prior studies show that the Bhutanese refugee community also experience poor mental health, high levels of depression, anxiety, PTSD, suicide, and substance abuse [13,14], largely due to cumulative migration disadvantages and lack of culturally responsive services [15,16].

The lack of access to digital technology is a serious issue in the Bhutanese community, and the pandemic and limits on direct care deepened the digital divide emphasizing already pervasive racial and cultural disparities especially among immigrant and refugee communities. Anecdotal data from community members reveal that Bhutanese households often lack internet access; as well as affordable data plans and efficient digital devices which impede protective strategies (remote work options, telehealth, and online access) and renders social activities such as online worship services and cultural celebration nearly impossible [5].

### 1.2. Critical Public Health Needs

In a recent methods conference, our team shared the conceptualization, implementation, and the quantitative findings from our digital divide research project. The conference challenged participants to explore the role quantitative research can play in capacity building and the quest to define social justice outcomes [17,18]. Conference participants addressed a myriad of topics ranging from measurement bias to equity issues in programming and design as well as the role engaged, coproduction and mixed methods research hold in the effort to represent the underrepresented, develop pathways for inclusion in higher education, research, and the study of health inequalities in general.

We shared our work as an important response to the conference planners’ challenges. Our project, a collaboration between the Bhutanese refugee community, anthropologists, and social workers is an opportunity to identify new pathways to transdisciplinary research and foster coproduction and unity as together we commit to equity and define sustainable solutions to real-world challenges.

Guided by the transformative principles of shared learning, equitable partnership, and participation [19], our investigation focused on defining the dynamic and complex structure of the digital divide and its impact on the Bhutanese community with particular attention to the disruption caused by COVID-19. Three key questions drove our efforts. First, we asked, what is the digital divide as it relates to the Bhutanese community? We sought to move beyond the assumption that the digital divide was geographic and defined by the technological gap that limits access to information in resettlement spaces. Second, we asked how the digital divide impacts the Bhutanese community, access to healthcare and information associated with the pandemic, including access to family support, personal protective equipment (PPE) and medical care. Third, we created a framework that looked past the pandemic to foster sustainable solutions that build toward a more equitable future for the community and create a safe space for Bhutanese refugees and their children to access education and educational opportunities and join as partners (rather than subjects) in ongoing research [20,21].

Our first conversations in Spring 2020 were key to understanding the impacts of the COVID-19 pandemic and the realization that uneven access to information was limiting the ability of the Bhutanese refugee community and its leadership to effectively respond. This did not mean that there was no response, and in point of fact, the BRAVE (Bhutanese Response Assistance Volunteer Effort) project established a voluntary network to act as “a go-to resource for families in need of COVID-19 screenings and testing, educational materials, PPE and other essential supplies, food, medicine, economic stabilization support, and vaccines” (see their web page at https://www.bccoh.org/brave.html, accessed on 1 October 2020) in approximately 13 cities around the nation. Nevertheless, challenges remained, and we focused our efforts on how we might best work together to build upon efforts to affect a more pro-active response to the challenges of COVID-19 and at the same time acknowledge and address the social inequalities that characterize minority healthcare. Additionally, we wanted to move beyond the pandemic and look toward a future that would include researchers and community members working together to foster the flow of information and communication, mentor students and co-create programs for the community to document itself [22].

## 2. Materials and Methods

Our mixed methods approach to data collection was reviewed and determined exempt by the Office of Responsible Research Practices (ORRP-OSU Study ID 2021E9088) and is a part of the transformative paradigm we developed to foster coproduction and engagement between Bhutanese community members and academicians (see Figure 1). Using a three-phased explanatory sequential mixed methods research design [23], we integrated (1) geographic information systems to inform (2) quantitative and (3) qualitative data collection. While this paper is based upon quantitative data and the open-ended responses to our survey, our larger study combines mapping, quantitative data, surveys, and ethnography to identify physical, social and health inequalities around the digital divide.

### Research Approach

Data collection included a community survey with 493 respondents. While some individuals accessed a Qualtrics survey online, many were conducted in person by trained community members (many of whom were Bhutanese undergraduates at a local university). Qualitative data emerged from open-ended questions that were embedded in the survey.

Field interviewers worked directly with our team members to identify respondents. Many of the respondents we identified were related to interviewers, helping to establish secure, disease free “bubbles” for research. It was critical that our respondents were safe and comfortable, and trusted the fieldworkers who engaged them. Familiarity, social ties (and sometime family connections) supported engagement during some of the most dangerous months of the pandemic in the spring of 2020. Working directly with community members who lacked access to broadband services and did not have technologies that would support connecting to the internet was critical as we captured the reality of daily life and documented the physical, social, and cultural characteristics of the digital divide.

The online survey, while only open to people with digital access and therefore problematic when studying the digital divide, allowed us to learn about the challenges and confronted users and disrupted access. We also were able to follow the various strategies that Bhutanese refugees used to manage broadband including a reliance on internet signals that originated outside of their homes. Finally, the qualitative data that typically clusters the Bhutanese with other South Asian populations (including the MASALA or Mediators of Atherosclerosis in South Asian Living in America study) created an effective sounding board for comparison and to highlight the unique qualities and challenges that face the community and may not always be obvious when South Asians from different sending countries, nations and ethnicities are grouped together [24]. Combined, these data sets allowed us to develop a clearer understanding of the challenges the digital divide creates for community members (see Table 1). What follows are descriptive results from our quantitative data, as well as univariate and bivariate analysis using SPSS version 22 (IBM, Armonk, NY, USA).

## 3. Results

The in-person and online surveys were combined to provide results from a sample of 493 respondents after all the missing data were analyzed (see Table 1).

Broadband access, while imperfect, is available to the community and more than one service provider is typically available across the area. About 95% of the Bhutanese surveyed for this study have access to the internet and can “surf the web” (see Figure 2).

Nevertheless, this high level of access does not equal equitable engagement. In other words, access to service, while important for understanding the reach and quality of broadband [25] does not capture access by the community and does not identify other factors that can limit access [26].

## 4. Discussion

While access to services increases with competition between providers and declining broadband costs [9,26], education and skill proved critical to a person’s comfort level and engagement with the internet. Of the Bhutanese community member surveyed, those with a college education or working as professionals reported they were skilled, very skilled, or extremely skilled when asked to rank their ability to work with internet technology (Table 2).

Individuals who lacked formal education or had completed only primary school and/or high school were not nearly as likely to report they possessed the skills necessary to use the internet, and for these individuals and their families, access was not guaranteed even if broadband was available. Education really does matter, as does age. Only 4 individuals with no formal schooling ranked themselves as very skilled or extremely skilled when working online (less than 1% of the respondents). While 39 individuals with high school diplomas classified themselves as very or extremely skilled (just over 8% of the respondents), younger Bhutanese (18–24) were among the most active and skilled users in the community (see Table 3). The problem was more than defining access and the presence of broadband providers in a community, rather access was predicated on education, age and the level of comfort as well as preparation. Expanding the reach of a signal or underwriting the extension of a signal in a community or neighborhood will not, by itself, solve the gap in service, access and/or use of broadband.

In other words, the gap separating the Bhutanese community from the internet goes beyond access to technology and comfort and builds upon a history of inequality, mistrust, poor translation, problematic expectations, and populational studies that lump groups together according to geographies while ignoring deeply held intragroup differences that are sometimes associated with violent historical periods (see Figure 3).

Inequality confronts the Bhutanese in Columbus daily and is founded in the historical experiences of the community and systematic discrimination in Bhutan, the refugee experiences of its members in Nepal and India, and the complicated negotiation of settlement in the US that brought the Bhutanese to urban centers throughout the Midwest [27].

The Bhutanese were stripped of their citizenship, rendered homeless and forcibly expelled from Bhutan in the late 1980s and early 1990s [28]. Forcibly removed from their homeland by the Bhutanese government, the population fled to Nepal and India at which point they were rendered pawns with few rights or opportunities. Nepal barred the assimilation and eventual citizenship for refugees in a process that continues to hinder the opportunities and outlook for the refugees who remain in camps [29,30]. UNHCR and US governmental settlement support marked a turning point for the Bhutanese community. While resettlement assistance supported the community, particularly as it met the immediate needs of its members; the health-related costs of past abuse and historical violence that confronted the Bhutanese during their expulsion continue to impact community members in the present [31,32,33]. The inequality that confronts the Bhutanese is complicated by mistrust that surrounds programming that is plagued by poor translation. One contact described state provided translations as someone running English text through an untested Nepali filter and sharing the results with no review or discussion. Additionally, there is a disconnect between what the community may require and what is provided. Poor translations, limited support, and a sense that the Bhutanese should be grateful for what they have rather than what they want or need, creates the expectation that the community is ungrateful, demanding, and problematic. These expectations are only complicated when populational studies lump the Bhutanese with other South Asians, and in particular Asian Indians, who are healthier, successful, and better integrated in the US educational system.

To understand what the digital divide is for the community we looked beyond technology and geography and consider socio-cultural factors that might contribute to isolating the Bhutanese. While the Bhutanese can access the internet and find information on a range of issue through the BRAVE project, most Bhutanese do not use the internet for information or to connect with programming. Rather they use the internet to contact and stay in touch with family and friends (see Figure 4). In fact, while nearly 83% of our respondents used the internet to engage with family and friends, and another 68% visited social media sites (including Facebook, the BRAVE project’s home), only 20% visited online news and informational websites that were not social media (Facebook, Twitter, and the like).

Language is a particularly problematic challenge for community members. While many Bhutanese can communicate in English, less than 25% describe themselves as excellent speakers (see Figure 5). Nuances and metaphors can be a challenge as can highly technical speech and medical speech, this challenge is complicated when the native speaker is already coping with discriminatory practices around work and community engagement [16,34,35].

The lack of programming in Nepali is challenging for the community at large, but most of all for the 27% of respondents who have limited English proficiency and difficulty with speaking and reading. Additionally, many Bhutanese, while native Nepali speakers, are not fully literate in Nepali and can have difficulty following Nepali script when it appears on the internet. In fact, some of these speakers will use Nepali for verbal communication but prefer a Latin alphabet when writing.

Though the majority of the Bhutanese community can access the internet, many overlook it as a resource or as a tool that supports access to information on health and wellbeing. This becomes clear when we asked respondents about their access and use of telehealth and mental health services (Figure 6). The pandemic and threat of COVID-19 has driven increasing reliance on telehealth services [36], yet satisfaction, trust and engagement with non-English speakers and refugee communities remains problematic at best [37]. While respondents recognized the importance of telehealth, nearly 73% of them had not used telehealth services to access health care or mental health care.

While telehealth is a critical tool in care and its popularity has increased [38], there remain challenges and patients continue to prefer in person appointments [39]. This is especially true for minority and refugee patients who are dealing with limited digital and health literacy [40,41].

Limited health literacy and problematic linguistic skills as well as their historical marginality leave the Bhutanese at a disadvantage when it comes to care and access to health-related information, including information on COVID-19 (see Figure 6 and Figure 7). More than a challenge to access, the lack of engagement creates the misbelief that all is well within the Bhutanese community. The challenge is one that is noted by researchers focused on inequalities in the healthcare systems globally [42,43]. These challenges are exacerbated when a refugee group, like the Bhutanese, experience deprivation after deprivation as they were expelled by one state and marginalized by another. The limited role that telehealth plays for the Bhutanese community complicates access to critical health information. Limiting access to information can negatively impact Bhutanese children who may, for example, fall behind on mandated vaccines as their parents disregard on-line support and telehealth [44].

The costs that confront the community as resources are not accessed is also a concern. However, what if the burden of choice was not placed fully upon the Bhutanese—and pressure for them to use tele-health and digital mental health services as intended was swapped, and caretakers were tasked with providing accessible resources that the community might use with confidence? The challenge is increasing; and as we transition from a focus on the pandemic, other health concerns that were placed on the backburner have reappear. This is most clearly illustrated if we look at cardio-metabolic disease in the community [45]. The Bhutanese community is at high risk for cardiometabolic diseases, and about 72% of the population exhibit 1 or more comorbidities (see Figure 7). Nevertheless, community members, in general, have not used telehealth resources in response to this challenge. This is not simply a challenge to wellbeing, it puts individuals at higher risk for heart attack, stroke, diabetes, and liver disease.

## 5. Conclusions

Our results are critical to addressing health inequalities and emphasize three key findings. First, the digital divide is not simply a question of technology and access. The digital divide may begin in limits on technology and broadband access, but historical inequalities and the marginality that often follows are critical to outcomes as socio-cultural differences impact expectations for care and support. This approach, also defined as Participatory Action Research [46], engages with the community and its members who, as part of the research team, work to actively improve health and wellbeing.

Second, the complexity of the digital divide and the historical trauma and social marginalization that confront the Bhutanese community’s health and wellbeing. As we have demonstrated, even with the presence of effective internet technology, available bandwidth and providers, healthcare is not necessarily accessible. We need to engage with Bhutanese community members to make sure we are using the correct language, framing challenges in a way that takes it cue from practice and belief, and not assume there is a quick solution to any problem.

Third, if we are to reach a more equitable and just future, we must engage the communities we study to actively engage with us as partners in the co-production of research as well as the analysis of the response. This means that in the immediacy of a challenge like the pandemic, we work quickly with the community and with specific goals in mind. However, we also must remember that our concerns (academic concerns) are not necessarily shared by the community’s we seek to engage. Even when there is little urgency, it is critical that we engage, pay attention, and understand that the people we work with are critical as together we set the agenda for our projects. For us, this meant that we were focused on understanding the digital divide, and in response, increasing digital literacy and building upon the BRAVE program to promote engaged action that can foster a sense of worth and value. This includes establishing a recording studio for members of the community to document their community, mentoring, and defining opportunities for younger Bhutanese to succeed in high school, college, and post-graduate education; and teaming with doctors and healthcare providers in an effort to bridge the digital divide in real time and through programming that can highlight issues around mental health and wellbeing following a design that meets local expectations. Finally, we hope to establish a research review board, not to limit the research conducted in the community, but for the community to be aware of the investigations that are taking place and to ask that those investigators make a direct contribution to the Bhutanese community. Whether the contribution will take the form of a report, a presentation or an intervention remains to be determined. However, it is critical to continue community engagement and recognize a history of inequality will not be resolved quickly, and trust cannot be promised, it must be earned. While we cannot infer causality in our findings, we are dedicated to expanding the scope of our analysis and collect additional data, including objective measures of wellbeing, to foster both research and continue community engagement as together we address health inequalities and disparities.

## Figures and Tables

**Figure 1 ijerph-19-16854-f001:**
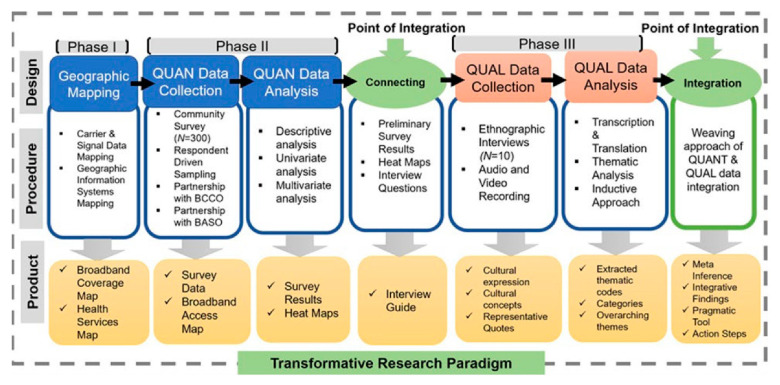
Transformative Research Paradigm.

**Figure 2 ijerph-19-16854-f002:**
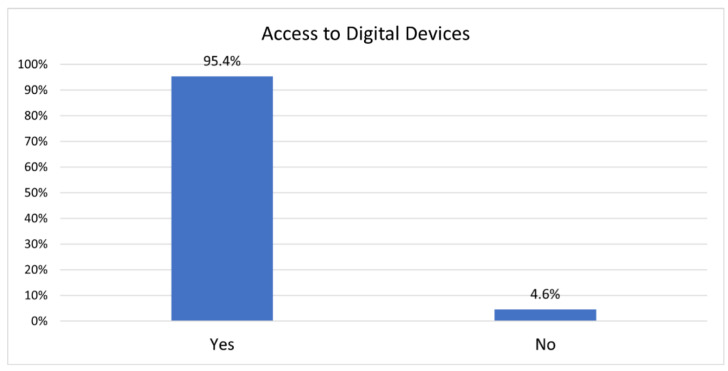
Participants’ access to digital devices.

**Figure 3 ijerph-19-16854-f003:**
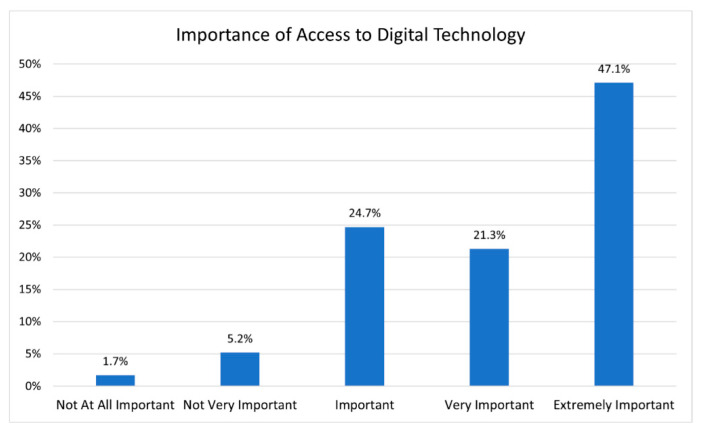
Importance of access to digital technology.

**Figure 4 ijerph-19-16854-f004:**
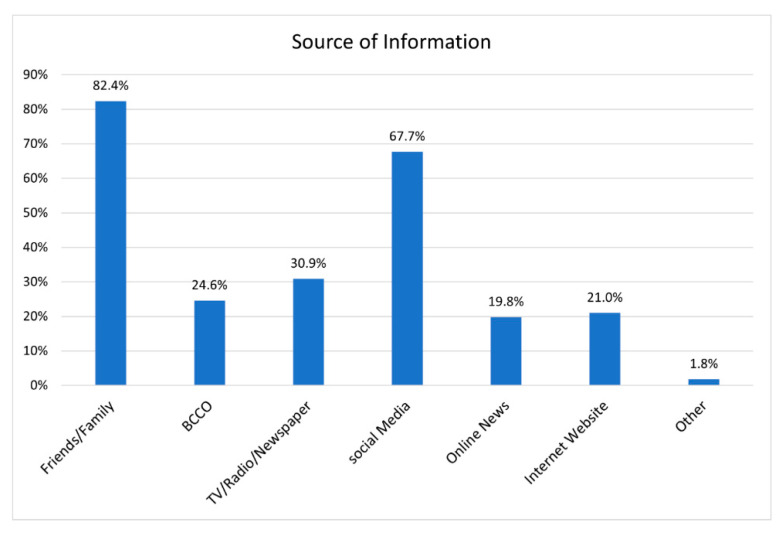
Source of information for participants.

**Figure 5 ijerph-19-16854-f005:**
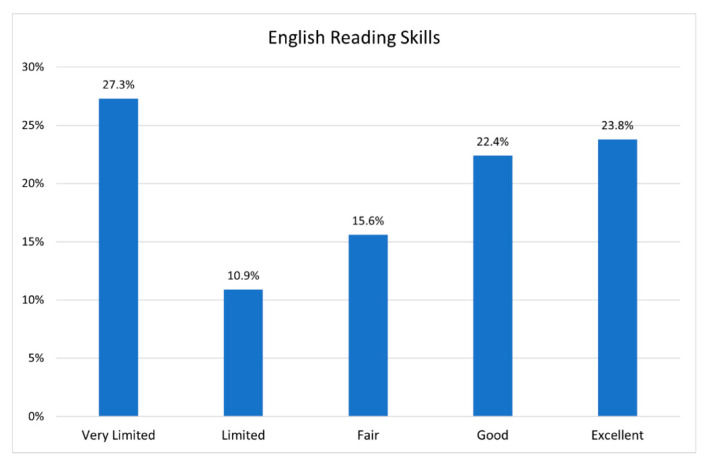
English reading skills of the participants.

**Figure 6 ijerph-19-16854-f006:**
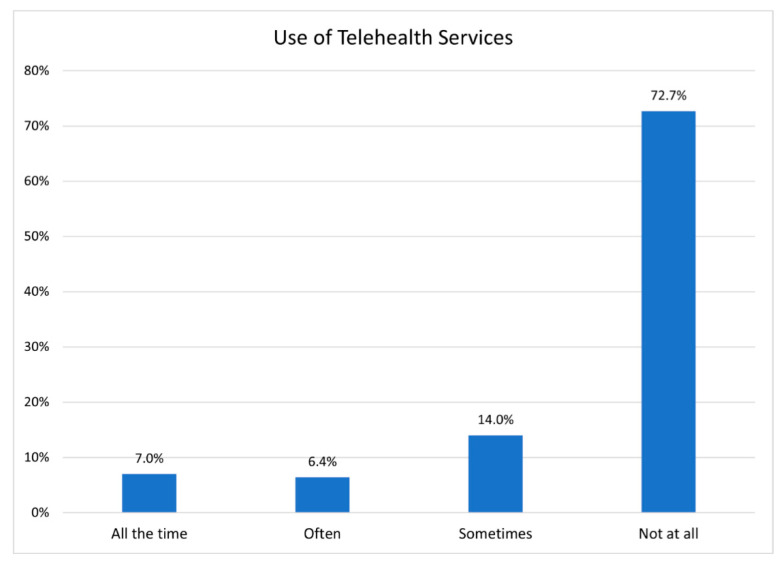
Use of Telehealth Services.

**Figure 7 ijerph-19-16854-f007:**
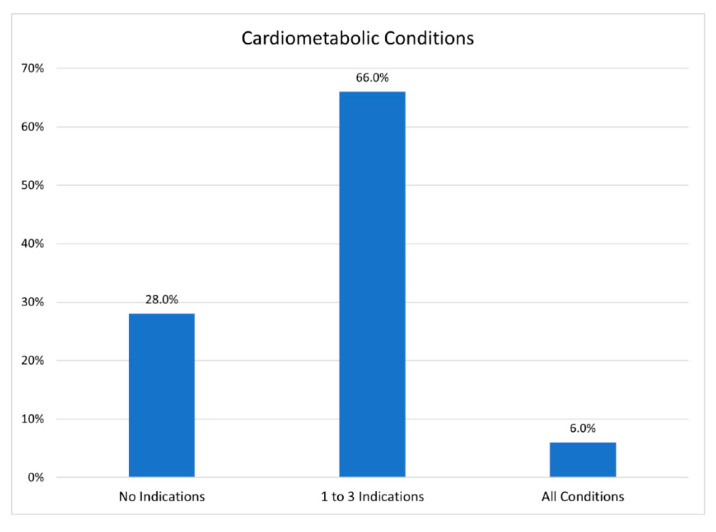
Participants’ Cardiometabolic Conditions.

**Table 1 ijerph-19-16854-t001:** Sociodemographic characteristics of the central Ohio Bhutanese community.

	N	%
Age group (N = 493)		
18–24 years	161	32.7
25–44 years	183	37.1
45–64 years	104	21.1
65 years and older	45	9.1
Sex (N = 495)		
Female	240	48.5
Male	255	51.5
Housing (N = 494)		
Own a house	341	69.0
Rent a house	48	9.7
Rent an apartment	90	18.2
Other	15	3.0
Place of birth (N = 475)		
Bhutan	270	56.8
Nepal	195	41.1
Other	9	1.9
Years of arrival in the US (N = 495)		
2002–2007	6	1.2
2008–2013	366	74.0
2014–2020	110	22.2
Citizenship Status (N = 487)		
Permanent Resident	167	34.3
US naturalized citizens	317	64.0
Other	3	0.6
Marital status (N = 491)		
Single	190	38.4
Married	270	54.5
Separated	6	1.2
Divorced	11	2.2
Widowed	14	2.8
Religion (N = 489)		
Hindu	383	78.3
Christian	57	11.7
Buddhist	31	6.3
Other	18	3.6
Ethnic Group (N = 491)		
Dalit	72	14.5
Janajati	76	15.3
Chhetri & Brahmin	307	62.5
Other	12	2.4
Prefer not to answer	24	4.9
Language spoken at home (Multiple Choice)		
Nepali	489	NA
English	173	NA
Hindi	37	NA
Education (N = 490)		
No formal education	120	24.5
Primary level	51	10.4
High school or GED	142	29.0
College	156	31.8
Professional degree	16	3.3
Other	5	1.0
Employment status (N = 483)		
Employed full-time	243	50.3
Employed part-time	70	14.5
Self-employed	5	1.0
Employed temporarily	6	1.2
Retired	17	3.5
Unemployed	131	27.1
Other	11	2.3
Access to digital device (N = 482)		
Yes	460	95.4
No	22	4.6
Annual family income (N = 463)		
Under $15,000	75	16.2
$15,000–24,999	80	17.3
$25,000–34,999	116	25.1
$35,000–49,999	82	17.7
$50,000–74,999	70	15.1
$75,000 and above	40	8.6

**Table 2 ijerph-19-16854-t002:** Digital technology skills by educational level of participants (N = 473).

	How Would You Rank Your Digital Technology Skills?				
	Extremely Skilled	Very Skilled	Skilled	Not Very Skilled	Not Skilled
	n (%)	n (%)	n (%)	n (%)	n (%)
Level of education					
No formal education	1 (0.9)	3 (2.6)	23 (19.8)	47 (40.5)	42 (89.4)
Primary level	1 (2.0)	1 (2.0)	17 (34.7)	29 (59.2)	1 (2.0)
High school or GED	11 (7.9)	28 (20.1)	71 (51.1)	27 (19.4)	2 (1.4)
College	49 (32.9)	54 (36.2)	42 (28.2)	2 (1.3)	2 (1.3)
Professional degree	4 (26.7)	6 (40.0)	5 (33.3)	0 (0)	0 (0)
Vocational or other	2 (40.0)	0 (0)	2 (40.0)	1 (20.0)	0 (0)

**Table 3 ijerph-19-16854-t003:** Cross-tabulations by age and digital technology skills and usage.

	How Would You Rank Your Digital Technology Skills?					Access Internet at Home			Access to Digital Device	
	Extremely Skilled	Very Skilled	Skilled	Not Very Skilled	Not Skilled	Never to Monthly	Weekly	Daily	No	Yes
	n (%)	n (%)	n (%)	n (%)	n (%)	n (%)	n (%)	n (%)	n (%)	n (%)
Age										
18–24 years	47 (69.1)	49 (53.3)	45 (28.3)	8 (7.6)	5 (10.6)	6 (17.1)	10 (31.2)	139 (34.0)	3 (13.6)	152 (33.2)
25–44 years	21 (30.9)	32 (34.8)	78 (49.1)	39 (37.1)	1 (2.1)	11 (31.4)	10 (31.2)	153 (37.4)	4 (18.2)	172 (37.6)
45–64 years	0 (0)	11 (11.9)	31 (19.5)	43 (41.0)	16 (34.0)	5 (14.3)	6 (18.8)	92 (22.5)	2 (9.1)	102 (22.3)
65 years and older	0 (0)	0 (0)	5 (3.1)	15 (14.3)	25 (53.2)	13 (37.1)	6 (18.8)	25 (6.1)	13 (59.1)	32 (6.9)

## Data Availability

The data presented in this study are available on request from the corresponding author. An earlier version of this article was presented at QMC22 (Qualitative Methodology Conference 2022, The Ohio State University, Columbus, Ohio, 2 June 2022).

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
