# Peer review of "Exploring the Digital Divide among the Bhutanese Refugee Community during COVID-19: Engaged Research in Action"

_ijerph, 2022, doi:10.3390/ijerph192416854_

Round 1

Reviewer 1 Report

This paper is interesting in that it  challenges us to examine the complexities surrounding the issue of digital divide for resettled refugees as as well as expanding our understanding of the dynamics and elements of what constitutes the digital divide, which is applicable when considering the experiences of other vulnerable, marginalized, and oppressed persons groups, communities, and cultures.  

The authors primarily use frequency distributions to support their thesis and answer the questions posed in the research. Did the authors perform any other types of descriptive statistics or inferential statistics other than that could be added here to answer the research questions. For instance, a cross tabulation of importance of access by age group may further provide supporting evidence. 

 The authors state " And while we can point toward the support by the UNHCR and the US as a turning point for the population; settlement assistance has focused on meeting the their immediate and basic needs and does not acknowledge past abuses, violence or the hopes and dreams for the future that  continue to drive refugees to seek opportunities (Pulla and Dhungel 2016; Rajesh 2003; and see the discussion in Ramsay 2017)." (line 216-221).    This statement should be reviewed as to its accuracy that "settlement assistance does not  acknowledge past abuses, violence ....".  No doubt this may be occurring, but perhaps was prevalent in the past as indicated by the citations used,  but is not necessarily characteristic of all practices or resettlement organizations as the statement implies or can be interpreted to be stating.  

The authors should clarify the following statements:

The Bhutanese are not ignoring the internet; rather, they are not in a place where they can use it. (line 257)

The Bhutanese do not access telehealth. (line 270)

Furthermore, the reticence that many Bhutanese feel toward telehealth extends beyond the individual to include members of their families and especially children who may fall behind on mandated vaccines among  other things (and see Tankwanchi, et al. 2021).( Line 278)

Accordingt to the data presented by the authors, the Bhutanese do use the internet for communicating with family, etc.  The authors should explain or define what they mean by "not in a place where they can use it"

Do the authors have data that they can include to indicate that the Bhutanese are/were offered telehealth during the pandemic, were reticent about it, and elected not to use it.  There were a number of systemic and structural barriers to using telehealth, which made access difficult for everyone.   The authors do sort of get at this starting with line 282.

The authors state" the complexity of the divide and the historical processes, social practices and cultural beliefs that define that complexity, must be understood if we are to address the 302 impacts of the divide on community health and wellbeing." ( Line 301-303).   It would seem reasonable for the authors to add statement to this regarding the role oppression, historical trauma, and marginalization figure into the complexity of the divide--  or are "historical processes, social practices"  includes these elements? 

In the conclusion section, the authors state that " if we are to reach a more equitable and just future, we must engage the communities we study and ask them to be partners in the co-production of research as well as the analysis of the response. " (line 308-310).  This would be a place for the authors to discuss the value and preference for participatory action research (PAR) and its methods which it appears they utilized in the course of conducting this study.

Reviewer 2 Report

Thank you for the opportunity to review the article "Exploring the digital divide among the Bhutanese Refugee Community during COVID-19: Engaged Research in Action". The article is engaging, exploring how a refugee community accessed healthcare and information associated with pandemics during the COVID-19 pandemic.

The theoretical background is appropriate for this subject; the authors describe the theoretical perspectives used in studying the digital divide in the first section of the article. Also, the context overview – the description of the Bhutanese Refugee Community it's appreciated.   

Overall, the subject is correctly attributed to the Health Behavior, Chronic Disease and Health Promotion section of the International Journal of Environmental Research and Public Health journal, special issue "Reducing Health Disparities and Promoting Healthy Youth, Families, and Communities" by studying a contemporary situation and its impact on an at-risk community. 

It is important to emphasize that the article is well-structured, and the results are clearly presented. 

However, there are some suggestions that the authors could introduce to improve the overall quality of the article.  

·      First of all, I recommend formatting the figures in which the statistical results are presented in the same dimension for a more accurate visual framing. Line 43: after Spielberger, the authors should put a reference according to the citation style.

·      While I appreciate the methodological complexity proposed by the authors, I believe that the methods applied should be described more fully in the methods and research approach section - what was the survey sample, and how were the methods applied ...there are some gaps regarding the procedure by which the methods were applied.

·      The results are presented synthetically; I would like, if possible, a more extended presentation of them that would make a more substantial reference to the three questions the authors used as premises. 

·      What are the limits of the research? Research limitations and constraints should be presented in the conclusions section.
